# Modeling Real-Time Interactive Conversations as Timed Diarized Transcripts

## Abstract

Chatbots built upon language models have exploded in popularity, but they have largely been limited to synchronous, turn-by-turn dialogues. In this paper we present a simple yet general method to simulate real-time interactive conversations using pretrained text-only language models, by modeling *timed diarized transcripts* and decoding them with *causal rejection sampling*. We demonstrate the promise of this method with two case studies: instant messenger dialogues and spoken conversations, which require generation at about 30 tok/s and 20 tok/s respectively to maintain real-time interactivity. These capabilities can be added into language models using relatively little data and run on commodity hardware.

## 1 Introduction

Chatbots built upon language models have exploded in popularity, but their interaction model is extremely limited: the user and the system take turns writing messages, where the system waits until the user finishes their message to respond then responds instantly and uninterruptibly. Extensions to support audio have used speech to text and text to speech to eliminate the need for typing and reading the screen (OpenAI, 2023), but the constraints of the interaction model have remained the same.

In this paper we present a simple method to simulate real-time interactive conversations using pretrained text-only language models. Namely: model *timed diarized transcripts*—i.e., sequences of [timestamp, speaker id, message]—at the desired granularity, and then decode these transcripts with *causal rejection sampling*—i.e., sample a continuation that will be finalized at the predicted timestamp, and if there is intervening user input before the timestamp, reject the planned continuation (to the extent that its probability under the model has changed) and resample a new one. This method is naturally sparse over time and number of speakers, scaling computation with the amount of content being actively produced at each moment. It is also quite general; in principle, it can also be applied to any task involving timed sequences of events, from time series forecasting to applications in gaming.

We demonstrate the promise of this method with case studies in two domains. First, we use the instant messenger chat history between the first authors to train a real-time interactive asynchronous text dialogue model. Second, we use public speech datasets with diarized transcripts to train a real-time spoken conversation model, cascaded through word-level speech to text and text to speech models. Here there is an additional complication in that real-time streaming speech to text systems are unstable, i.e., predictions may change in light of future context. We address this with *retconning*, i.e., revising the user's input history but keeping any already finalized system outputs.

We evaluate these embodiments of our method with respect to performance (properties of the control token format and of our proof-of-concept implementation) and quality (test perplexity, offline human ratings, and online human ratings)—across finetuned models from 160M to 12B parameters. For the offline human rating setting only, we also use long in-context learning to test larger pretrained models available by API. In order to maintain real-time interactivity, generation needs to be about 28 tokens per second for the instant messenger use case and 22 tok/s for spoken conversations, which are easy to achieve on a single A100 at our model scales. We find that, predictably, better pretrained models lead to better results, though there is still obvious room for improvement with dataset/model scale.

We publicly release our code (and some demo videos) at this link. We hope that these proofs of concept spark the imagination and show that language models can easily be adapted to new real-time interaction modes.

## 2   METHOD

We model *timed diarized transcripts* using causally masked (decoder-only) language models. Given a sequence of events $e_i$, where each event $e_i$ consists of a timestamp $t_i$ (*timed*), a speaker id $s_i$ (*diarized*), and a message $m_i$ (*transcript*), we model $p(e_i|e_1, ..., e_{i-1})$. In practice, this function decomposes into $p(t_i|e_1, ..., e_{i-1})$, $p(s_i|e_1, ..., e_{i-1}, t_i)$, and $p(m_i|e_1, ..., e_{i-1}, t_i, s_i)$, or even more granular distributions if these components are represented as multiple tokens. By modeling events sparsely over time, we are able to sample transcripts with computation proportional to the number/complexity of the events, rather than the time duration.

In order to make this model interactive, we use *causal rejection sampling*. We pick a particular speaker id $S$ to represent the user and sample candidates $\hat{e}_i \sim p(e_i|e_1, ..., e_{i-1})$, where we interpret the timestamps $t$ within these events with respect to the current real time. If an input from the user $(S, T, M)$ interrupts before the timestamp $\hat{t}_i$ is reached, we reject the candidate $\hat{e}_i$ and sample a new candidate $\hat{e}_{i+1} \sim p(e_{i+1}|e_1, ..., e_{i-1}, e_i = (S, T, M))$. If no such interruption occurs before $\hat{t}_i$, there are two possibilities: If the speaker id $\hat{s}_i$ within $\hat{e}_i$ is not $S$, we accept the message candidate $\hat{m}_i$, emit it to the user, then sample $\hat{e}_{i+1}$, etc. If $\hat{s}_i$ is $S$, then we resample $\hat{e}'_i \sim p(e_i|e_1, ..., e_{i-1}, t_i \geq \hat{t}_i)$.

Because it takes some amount of time $t_{latency}$ (varying with message length) to execute the model and sample from $p(e_i|...)$, if the user repeatedly provides input less than $t_{latency}$ before the predicted timestamps $\hat{t}_i$, the model will be starved and unable to generate any acceptable events. We provide two modifications to mitigate recomputation from user interruption:

First, we enforce a hard lower bound on the model's generation bandwidth by stipulating that if the user input comes within $t_{react}$ of $\hat{t}_i$, we accept $\hat{e}_i$ as a candidate for $\hat{e}_{i+1}$. The relationship between $t_{latency}$ and $t_{react}$ determines whether the model can maintain real-time interactivity in the worst case. We do not expect moderate $t_{react}$ to harm generation quality too much because a human reaction time of approximately 150-200 ms (Thompson et al.; Jain et al.) should be reflected in the underlying causal structure of human training data.

Second, we reduce the average amount of recomputation by integrating speculative decoding (Leviathan et al., 2023; Chen et al., 2023). Rather than discard the candidate $\hat{e}_i$ unconditionally upon user interruption, we treat it as a draft for the new generation, rejecting and resampling based on the closeness of $p(e_i = \hat{e}_i|e_1, ..., e_{i-1}, t_i \geq T)$ and $p(e_{i+1} = \hat{e}_i|e_1, ..., e_{i-1}, e_i = (T, S, M))$. Note that this is different from traditional speculative decoding, where a smaller model *for the same distribution* drafts a candidate;[1] the use of different prompts under the same model resembles classifier-free guidance (Ho & Salimans, 2022; Sanchez et al., 2023). Like with $t_{react}$, we expect this to work to the extent that there is a looseness in the causal dependencies of nearby messages from different parties.[2]

See Algorithm 1 for a formal description of causal rejection sampling (speculative decoding omitted for clarity; see Appendix A for the full version), or see our code at this link.

We now present two case studies demonstrating how this method can be applied to different domains: instant messenger dialogues and spoken conversations.

### 2.1   INSTANT MESSENGER DIALOGUES

The method as described above can be applied to instant messenger dialogues with minimal modifications. We use as our domain 9 years of instant messenger history between the first authors. This means we are not just modeling the evolution of synchronous conversations where both participants are actively engaged, but asynchronous conversations where participants may be offline and where the date/time may influence the content of the conversation. Instant messenger conversations can be highly multimodal, in particular with audio, images, and hyperlinks; we consider only text and leave multimodality to future work.

---

[1]The traditional kind of speculative decoding could also be used to speed up the initial autoregressive candidate generation; we omit this for simplicity.

[2]You can also trade off between potentially wasted computation and interactivity by sampling the timestamp first and waiting until it approaches to generate the rest of the message, vs. sampling multiple sequential event candidates ahead of time.

---

**Algorithm 1** Causal rejection sampling (without speculative decoding)

---

$i \leftarrow 0$           ▷ current event index
$e \leftarrow []$           ▷ event history
$c \leftarrow (\varnothing, \varnothing, \varnothing)$           ▷ candidate for the next message
**while** true
    $i \leftarrow i + 1$
    **try**
        $(\hat{t}, \hat{s}, \hat{m}) \leftarrow c$
        **if** $\hat{t}$ is $\varnothing$
            $c \leftarrow (\hat{t}, \hat{s}, \hat{m}) \sim p(e_i | e_1, ..., e_{i-1}, t_i \geq t_{cur})$
        **wait** until $\hat{t}$
        $t_{cur} \leftarrow \hat{t}$
        **if** $\hat{s}$ is $S$
            $c \leftarrow (\varnothing, \varnothing, \varnothing)$
            $i \leftarrow i - 1$
            **continue**
    **catch** user input $(T, S, M)$
        $e_i \leftarrow (T, S, M)$
        $t_{cur} \leftarrow T$
        $(\hat{t}, \hat{s}, \hat{m}) \leftarrow c$
        **if** $\hat{s} = S$ or $\hat{t} + t_{react} < T$
            $c \leftarrow (\varnothing, \varnothing, \varnothing)$
        **continue**
    $e_i \leftarrow c$
    **emit** $c$
    $c \leftarrow (\varnothing, \varnothing, \varnothing)$

---

In the notation from above, we instantiate $t$ with the message's calendar date/time (down to decisecond granularity), $s$ with an id representing the message sender (one of the two authors), and $m$ with the message plaintext (terminated by an "end of message" token). As a sequence length optimization, when prefixes of the timestamp are repeated in consecutive messages, we omit them. We design the control format to be prefix-free so that it can be interpreted without lookahead while decoding; this means that control tokens can decoded in a structured way (including that time only flows forward) by appropriately filtering and renormalizing the next token vocabulary. See Figure 1a for a specification of the format and Figure 1b for an example of what preprocessed data looks like.

## 2.2 SPOKEN CONVERSATIONS

We also apply our general method to timed diarized word-level automatic speech recognition (ASR) transcripts. By cascading input through speech-to-text and output through text-to-speech, we can simulate spoken conversations. Note that—like cascaded approaches in general—this has the obvious limitation that it bottlenecks the input and output through text, stripping away aspects of speech like tone and introducing errors from intermediate models. While there exist off-the-shelf streaming speech-to-text models that output word-level timestamps, we are not aware of any text-to-speech models (streaming or otherwise) that accept them as input: the closest is incremental text-to-speech (Ma et al., 2020a). This limits our ability to generate natural-sounding speech; we use word-level text to speech invoked at the specified timestamps and consider this out of scope.

There is an additional complication due to the use of streaming speech-to-text models: these models are able to achieve low latency because they output preliminary transcriptions that may change in light of future input and are only finalized some time later. This means that not only can the user's input interrupt the model's candidate generation, but the input can retroactively change after a candidate has been generated, accepted, and spoken out.

We address this with *retconning*, i.e., when the speech-to-text model's prediction for the input changes, we replace the old prediction with the new one in the transcript prefix, without changing

$[[[[[[year?$ ' ' $month]?$ $day$ ' ' $wday]?$ '+' $hr]?$ ':' $min]?$ ';' $sec]?$ '.' $dsec$ $speaker$ $message$ <eom>

$year$: year in YYYY format (2015, 2016, ...)
$month$: full month name (January, February, ...)
$day$: date in DD format (01, ..., 31)
$wday$: day of the week (M, Tu, W, ...)
$hr$: 24-hour time in HH format (00, ..., 23)
$min$: minute in MM format (00, ..., 59)
$sec$: second in SS format (00, .., 59)
$dsec$: decisecond in D format (0, ..., 9)
$speaker$: message sender id (A | B)
$message$: plaintext message

(a) **Control token format.** "?" denotes an optional element. In brief: the format consists of the speaker id (omitted when matching the previous message), then the timestamp (prefixes omitted when matching the previous message), then the message itself. We use distinct separators ('+', ';', '.') between digit fields to distinguish them while decoding without lookahead, while remaining relatively tokenizer-agnostic. This format could be further optimized given a fixed vocabulary.

```
2024Feburary28W+22:32;13.8Bgetting
some cuda device error
though<eom>

;18.4Bthis is what I get for
developing on cpu...<eom>

;45.2Aone sec I'm running<eom>

33;03.6BI was also in the
middle of editing it so it's
not working too<eom>

34;15.4Bnvm fixed<eom>
```

(b) **Example of a formatted chat excerpt.** Newlines added for readability only; messages may include newlines in their plaintext, so <eom> is a distinct token absent in our training data.

Figure 1: **Formatting for the instant messenger case study.**

any model generations that were accepted after that point. More formally, if we have sampled $\hat{e}_j \sim p(e_j | e_1, ..., e_i, ..., e_{j-1})$ and the user interrupts with a revision $e_i'$, we reject $\hat{e}_j$ (subject to the $t_{react}$ window and speculation described above) and resample $\hat{e}_j' \sim p(e_j | e_1, ..., e_i', ..., e_{j-1})$. This should not have a significant impact on either performance or quality, since processing $n$ tokens in parallel is much faster than $n$ tokens sequentially, and because humans also reinterpret what they've already heard in light of new speech (which should be reflected in ground truth causal structure). See Appendix B for a more formal description of causal rejection sampling with retconning, or see our code at this link.

We use as our dataset 1000 hours of oral arguments before the U.S. Supreme Court (Team; Boyle, 2019). Court oral arguments are an interesting domain because they have many participants ($\sim 10$ per transcript) and are information dense, though they have longer conversation turns and fewer interruptions than typical conversations.

In the formal language from Section 2, we instantiate $t$ with the word's start timestamp modulo 10 seconds[3] (down to centisecond granularity), $s$ with an opaque identifier representing the speaker, and $m$ with the word plaintext (terminated by an "end of message" token). We omit the speaker id in repeated spans. See Figure 2a for a more complete description of the format and Figure 2b for an example of what preprocessed data looks like.[4]

## 3 EVALUATION

For both case studies we evaluate performance and quality. We finetune the following models: Pythia 160M, 1.4B, & 12B (Biderman et al., 2023), Gemma 2B (Team et al., 2024), and Llama 2 7B (Touvron et al., 2023); see Appendix C for details. Where possible, we also compare with in-context learning using state-of-the-art commercial language models: Claude 3 Sonnet (Anthropic) and GPT-4 Turbo (OpenAI). See Appendix D for details.

For performance, we report:

- *generation bandwidth* in tokens/second required to maintain real-time interactivity, scored on historical data

---

[3]This compromise reduces the number of tokens, at the expense of being able to model more than 10 seconds of silence.

[4]Note that for generality, the duration of each word should probably also be modelled. We omit it here because it would be discarded in our word-level text to speech step anyway.

$$sec\ dsec\ csec\ speaker\ word\ \texttt{<eom>}$$

*sec*: ones place of the timestamp in seconds (0, .., 9)
*dsec*: tenths place of the timestamp (0, ..., 9)
*csec*: hundredths place of the timestamp (0, ..., 9)
*speaker*: speaker id (A, B, ...)
*word*: plaintext word

(a) **Control token format.** "?" denotes an optional element. In brief: the format consists of the speaker id (omitted when matching the previous message), then the timestamp (prefixes omitted when matching the previous message), then the message itself.

```
055Aknock
079Aknock
154Bwho's
186Bthere
252Ainterrupting
316Acow
377Binterrupting
443Bcow
448Amoo
473Bwho
```

(b) **Example of a formatted word-level transcript** (out of domain). Newline serves as `<eom>`.

Figure 2: **Formatting for the spoken conversation case study.**

- *control token overhead ratio*, scored on historical data
- *speculation acceptance rate* as an average number and fraction of draft tokens, scored on historical data
- performance properties for the proof of concept implementation

For quality, we report:

- *document-level negative log likelihood (NLL)* on the held out test set (rather than token-level perplexity, to make comparisons meaningful across tokenizers)
- *offline human ratings*, i.e., a human ranks conversations that were generated by continuing a prefix from the test set noninteractively
- *online human ratings*, i.e., a human interacts with each model given a conversation prefix from the test set, and then ranks them
- statistics about the distribution of predicted time gaps, compared to historical data

For human rating settings, we use the same prefixes of 64 messages (∼1024 tokens) across all models. For the offline ratings, we also compare with the ground truth continuation. Note that while context lengths have recently made massive strides (128K for GPT-4 Turbo (OpenAI), >1M for Claude 3 Anthropic, and >10M for Gemini 1.5 (Reid et al., 2024)), they are still not long enough to fit our training sets (20.2M tokens of messenger history and 40.3M tokens of oral arguments) and usage is subject to rate limits. We therefore use only the most recent 16K tokens of history as context.

One of the first authors prepared the test harness; the other served as the rater. The human evaluation scores range from 0 to 6, where 0 is nonsensical and 6 is indistinguishable from real. These scores should only be used to judge relative quality and not quality in absolute.

## 3.1 INSTANT MESSENGER DIALOGUES

As our dataset we use 9 years of instant messenger conversation history between the first authors, totaling 37,649,697 characters across 1,393,508 text-based messages (we exclude messages from other modalities). We use the first 95% of the messages as the train set, the next 2.5% as a validation set, and the last 2.5% as a test set.

### 3.1.1 PERFORMANCE

See Figure 3 for details on the performance properties of our instant messenger control format. The highlights are: With $t_{react}$ = 200ms, the 99th percentile generation bandwidth required to maintain real-time interactivity is 28 tok/s, and the 99.9th percentile is 75 tok/s. This range is largely pathological cases like long pasted text. On average, the control-formatted token length is 3.2x the plaintext length (median 2.4x); speculative sampling saves an additional 11.02 draft tokens (69.5% of tokens) per interruption in Llama 2.

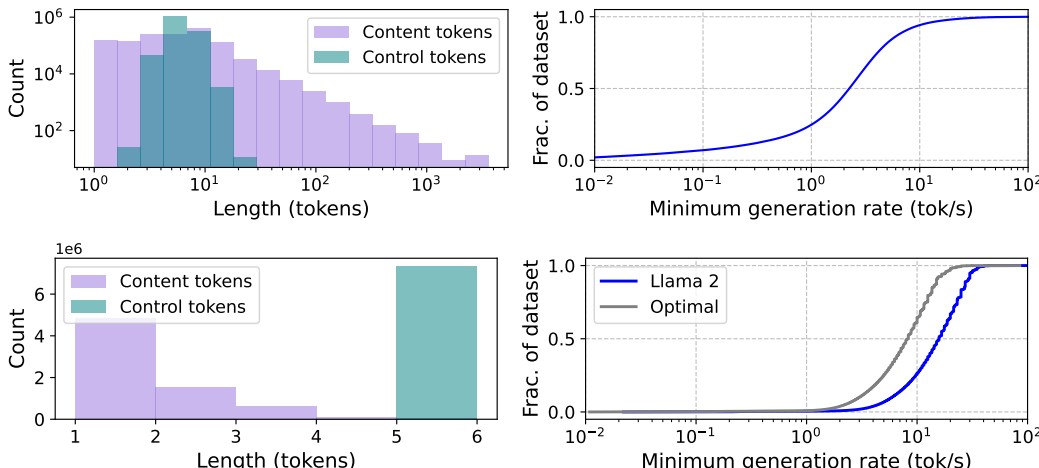

Figure 3: **Statistics about the overhead of our control formats for instant messenger dialogues (*top*) and spoken conversations (*bottom*), and the requirements to maintain real-time interactivity.** *Left:* Lengths (in Llama 2 tokens) of plaintext messages vs. control tokens for examples in the training set. *Right:* Fractions of the messages in the ground-truth dataset, including control tokens, that could be generated in real time for a given minimum generation rate, in tokens per second (again using the Llama 2 tokenizer). A message $m$ can be generated in real time if it can be generated in the time between the latest message outside of a short reaction window ($t_{react} = 200$ms) immediately before $m$, and $m$ itself. (We assume that for small $n$, the increase in cost for passing $n$ tokens through the network in parallel vs. 1 token is negligible, i.e. we are primarily modeling the cost of generating system responses, not ingesting user inputs.) For spoken conversations, we include performance figures for an optimized tokenizer which treats uses a single token for 3-digit timestamps.

In terms of our prototype: We interact with an A100 40GB server executing unquantized off-the-shelf model inference over `ssh`; this is more than sufficient to maintain real-time interactivity with all of our finetuned models. Communication latency is negligible, and the model checks for interruptions after generating each token (i.e., $\frac{1}{\# \text{tok/s}}$ latency).

### 3.1.2 QUALITY

See Table 1 for instant messenger quality results across models; see Appendix F for qualitative examples. The trends are unsurprising: better pretrained models achieve better perplexity and better human ratings, though still substantially worse than the ground truth. One exception is that API-based models with in-context learning mimic style worse than finetuned models, and sometimes fail completely due to refusals.

See Figure 4 for experiments comparing the distribution of predicted timestamps to the ground truth distribution.

We now describe some qualitative observations:

**Overpowering tone**  API-based models are tuned to have a particular voice, which bleeds through into the generated messages. So while the conversations are more coherent, they are usually easy to distinguish from the ground truth based on style cues alone. Claude 3 often refuses to perform the task when the chat history discusses politics.

**Speaker consistency**  The finetuned models sometimes struggle to maintain consistent identities for the speakers, mostly across conversations (e.g., one speaker talks about having a sister, when it is only the other speaker who has a sister) but sometimes also within conversations (i.e., a speaker appears to respond to itself).

**Promise as an evaluation for long context LLMs**  Instant messenger history continuation is a promising task for human evaluation of long in-context learning. Each message history is highly distinct, yet private and therefore guaranteed to be unleaked. While it is prohibitively time-consuming for a human rater to read extremely long prompts in general, if they are instead a participant in the

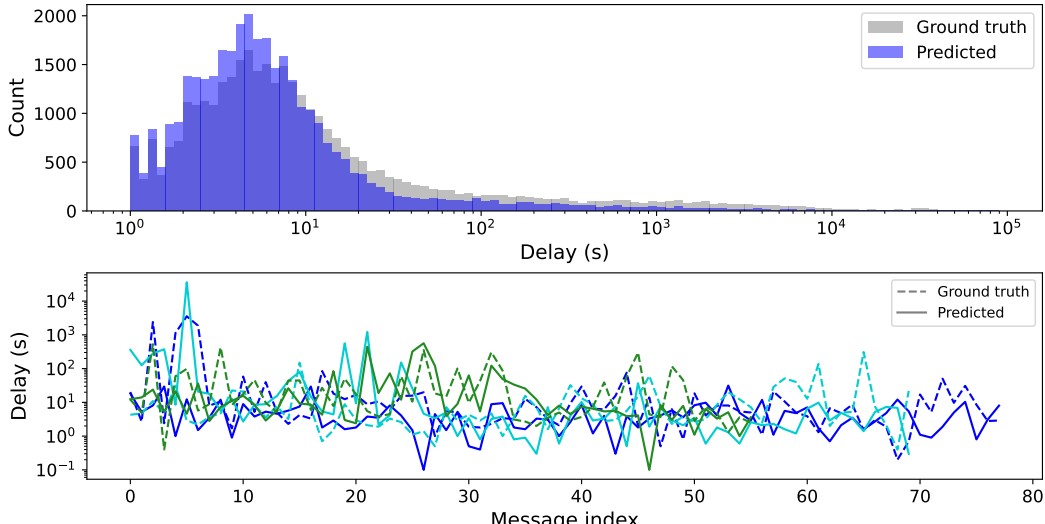

Figure 4: **Conversations generated by fine-tuned language models exhibit realistic message timings.** *Top:* Log-binned histogram of the delays (in seconds) between successive messages in 512 independent 1000-token conversations generated unconditionally by fine-tuned Llama 2 7B (temperature 1, top-p=0.95 (Holtzman et al., 2020)), compared to delays in a corresponding chunk of consecutive ground-truth messages of the same size sampled at random from the same month and year as the simulated ones. Mean conversation length is 73 messages. The empirical distributions are very similar (25-bin Kullback–Leibler divergence = 0.005), attributable to nucleus sampling. *Bottom*: Consecutive message delays for continuations of three randomly selected message history prefixes, ground truth (dotted) vs. predicted (solid). We do not expect these to perfectly match due to irreducible entropy, but the resemblance in trajectory shows that the model is not just learning first-order statistics.

original conversation, they are already deeply familiar with the content and can easily spot errors without additional effort.

### 3.2 SPOKEN CONVERSATIONS

As our training dataset, we use a random 1000-hour subset of cases argued before the U.S. Supreme Court, totaling 33,640,559 characters. We sample other cases into a ~350-hour val set and ~295-hour test set. We preprocess the data with WhisperX (Radford et al., 2022b; Bain et al., 2023), which supports timed diarized word-level ASR. Note that pseudolabeled diarized speech data tends to undercapture timestamp overlap across speakers (Liesenfeld et al., 2023), so this data may not reflect fine-grained turn-taking behavior. We lowercase and strip punctuation from the data to make the formatting consistent with streaming ASR.

#### 3.2.1 PERFORMANCE

See Figure 3 for more details on the performance properties of our spoken conversation control format. The highlights are: With $t_{react}$ = 200ms, the 99th percentile is 36 tok/s and 99.9th is 45 tok/s. On average, the control-formatted token length is 4.3x the plaintext length (median 5x). Note that this ratio is heavily dependent on the way the tokenizer handles digits; many modern tokenizers force individual digits to be separate tokens to improve arithmetic, but in this case, given enough data, 000-999 could reasonably be single tokens. We calculate the rates for this "optimized tokenizer": the 99th percentile is 22 tok/s and 99.9th is 30. On average, the control-formatted token length is 1.8x plaintext length (median 2.0x).

For our proof of concept implementation, we use Google Cloud streaming Speech-To-Text and Text-To-Speech APIs on the client, piped through an `ssh tty` as text to an A100 40GB server. We

| Instant messenger | NLL (↓) | Offline Human Ratings (↑) | | Online Human Ratings (↑) | |
|---|---|---|---|---|---|
| | | Consistency | Fidelity | Consistency | Fidelity |
| `Pythia 160M (ft)` | 3181 | 1.45 | 3.00 | 1.4 | 2.6 |
| `Pythia 1.4B (ft)` | 2397 | 2.55 | 3.65 | 3.4 | **4.8** |
| `Pythia 12B (ft)` | 2305 | 2.90 | 3.70 | 3.0 | 3.0 |
| `Gemma 2B (ft)` | 2376 | 2.95 | 3.65 | 2.8 | 3.2 |
| `Llama 2 7B (ft)` | **2179** | 3.90 | **4.40** | **3.8** | 4.2 |
| `Claude 3 Sonnet (icl)` | - | 1.85 (5.29) | 1.25 (3.57) | - | - |
| `GPT-4 Turbo (icl)` | - | **5.30** | 1.80 | - | - |
| *ground truth* | - | 5.95 | 6.00 | - | - |
| **Spoken conversations** | | Content | Timing | Content | Timing |
| `Pythia 160M (ft)` | 2261 | 0.8 | 1.4 | 0.6 | 0.4 |
| `Pythia 1.4B (ft)` | 1724 | 2.3 | 3.8 | 1.0 | 1.0 |
| `Pythia 12B (ft)` | 1661 | 3.1 | 3.8 | 1.6 | 1.8 |
| `Gemma 2B (ft)` | 1608 | 3.9 | 4.3 | 2.2 | 3.4 |
| `Llama 2 7B (ft)` | **1532** | 4.3 | **4.8** | **4.0** | **5.2** |
| `Claude 3 Sonnet (icl)` | - | 4.2 | 3.7 | - | - |
| `GPT-4 Turbo (icl)` | - | **5.0** | 3.8 | - | - |
| *ground truth* | - | 3.7 | 3.9 | - | - |

Table 1: **Instant messenger (*top*) and spoken conversation (*bottom*) quality scores.** `ft` = finetuned and `icl` = in-context learning. We compute negative log likelihood per document rather than averaged per token, so that it is comparable across vocabularies. Human ratings range from 0 (worst) to 6 (best). When relevant, we provide scores in parentheses with refusals filtered out. We rate *consistency* (how coherent the conversation is generally) and *fidelity* (how well the model mimics the authors specifically) for instant messenger, and *content* vs. *timing* for speech. See Appendix E for more details and experiments comparing the ground truth and predicted timestamp distributions.

measure the end-to-end latency of the former at about 500 ms (from word end to model input) and the latter at about 80 ms; on-device cascade and base models would likely have even lower latency.

### 3.2.2 QUALITY

See Table 1 for spoken conversation quality results across models; see Appendix F for qualitative examples. It is prohibitively time-consuming to read the entire context or each case, and the rater has some legal knowledge but is not an expert, so there may be more of a gap in content quality than is reflected by the scores. In the offline human rating setting, we play the transcripts aloud to judge timing, though with word-level text to speech it is difficult to judge the finer points. Like for instant messanger dialogues, better pretrained models tend to achieve better results. Llama 2 7B (`ft`) responds remarkably well to turn-taking in the online setting, though there is still obvious room for improvement in all regards.

## 4 RELATED WORK

We survey related work in three areas: text dialogues, spoken dialogues, and use of language models to model time broadly.

### 4.1 TEXT DIALOGUE MODELING

Modeling text dialogues is perhaps the founding problem of artificial intelligence: Turing's imitation game poses the challenge of distinguishing man from machine through turn-by-turn text dialogue (Turing, 1950). While timing is mentioned here (a model that responds too quickly could be distinguished from a human), the interaction model is limited. Since then there has been a wealth of work on dialogue systems (Ni et al., 2022), initially with complex rule-based methods (Weizenbaum, 1966) but shifting over time towards unified deep learning methods, culminating in Meena & LaMDA (Adiwardana et al., 2020; Thoppilan et al., 2022), the Blenderbot series (Roller et al., 2020; Komeili et al., 2021; Shuster et al., 2022), and of course the recent wave of chatbots such as

ChatGPT (Schulman et al., 2022), Gemini (Google, 2024), Copilot (Microsoft), Claude (Anthropic, 2023), Pi (Inflection), Coral (Cohere), HuggingChat (HuggingFace), etc. These chatbot works have primarily focused on basic, goal-directed conversational capabilities in the desired domains, which until recently has been very challenging, and less on the interaction model. Replika (Replika) and certain modes in Character.AI (character.ai) do allow multiple messages per conversation turn, but with undisclosed methods and unclear limitations.

CICERO (Bakhtin et al., 2022) studies Diplomacy, a political strategy game that involves instant messaging with other players in real time. The primary focus is on using dialogue paired with actions to achieve certain goals in the game, which implies the ability to imitate natural timing to avoid raising suspicion with human players. CICERO uses a chain of encoder-decoder models and heuristics to perform tasks such as predicting the next message time vs. content independently, and not all context is available to all models. Messages are rejected/resampled when user input causally intervenes on planned messages. Our work uses a simpler approach with a single transcript in a decoder-only model, which minimizes recomputation and makes all information available for all decisions; we further improve performance by using a reaction time window and causal speculative decoding.

The task of imitating specific people based on their digital footprint (for better or worse) has captured the popular imagination, featuring in shows like *Silicon Valley*, *Black Mirror* and *Westworld* and described with names like generative clones or ghosts in academic literature (Morris & Brubaker, 2024). Blog posts about finetuning LMs on personal chat histories are relatively common, but they either model timed transcipts noninteractively, or synchronous turn by turn conversations interactively (as a traditional chatbot). We are not aware of prior work that turns models of timed transcripts into interactive applications.

## 4.2 SPOKEN DIALOGUE MODELING

To go beyond manually crafted turn-taking heuristics for what is in generality an extremely complex task (Skantze, 2021), the main approach for generating spoken conversations has been direct audio modeling. dGSLM (Nguyen et al., 2022), AudioLM (Borsos et al., 2023), and SpiRit-LM (Nguyen et al., 2024) do this by modeling learned discrete tokens with autoregressive language models; the former models two streams of audio (dialogues), while the latter two model one. While the token modeling is causal, the tokenization is not, so these methods do not directly work for streaming generation. In concurrent work, GPT-4o (OpenAI, 2024) offers an "Advanced Voice" mode, but it does not offer full interactivity (*e.g.* while users can interrupt the model, it cannot interrupt users) and relies on undisclosed methods.

Discrete audio tokenization is generally performed at a fixed rate of $\sim$40-50 tok/s for a single audio stream, vs. $\sim$20 tok/s for our approach supporting arbitrary numbers of speakers.[5] This fits into the general pattern of cascaded vs. end-to-end models: cascaded models are generally more performant/require less data and therefore can be developed sooner using fewer resources, but they are eventually superseded by end-to-end models which can provide the optimal quality given sufficient resources.

Though not exactly dialogue, simultaneous translation often operates through a cascade of ASR and TTS, though timing information (besides the relative ordering of words in the source and target streams) is stripped away (Ren et al., 2020; Ma et al., 2020b).

## 4.3 TIME-AWARE LANGUAGE MODELS

There are many works that make language models aware of time in one sense or another. Even without special effort, language models learn latent representations of time to the extent that it helps explain the training distribution (Gurnee & Tegmark, 2024). The language model CTRL (Keskar et al., 2019) is conditioned on metadata about each document, which may include the publication date. Whisper (Radford et al., 2022a) and some other speech-to-text models predict timestamps as text. Park et al. (2023) lets loose generative agents in a virtual town environment, where they act on

---

[5]With that said, as with sparse vs. dense approaches generally, under extreme load the bandwidth required for sparse indexing over time may be higher than dense tokenization without indexing. And because our approach is sparse over time, it is more difficult to batch and has inconsistent load, which may be disadvantageous for bulk serving.

schedules in accordance with the virtual time. Language models have been used as the backbone for time series forecasting, whether pretrained (Das et al., 2024), finetuned (Jin et al., 2024), or zero-shot (Gruver et al., 2023), though here time is usually dense (proceeds at a fixed rate). We are not aware of works that model timestamps as text and interpret those timestamps as an input/output stream with respect to the real-world time.

## 5 CONCLUSION

In this paper, we presented a simple yet general method for simulating real-time interactive conversations using pretrained language models—modeling *timed diarized transcripts* and decoding with *causal rejection sampling*—situated in two use cases: instant messenger dialogues and spoken conversations. It is easy to imagine extensions such as multiple simultaneous conversations with one simulated individual (by adding conversation ids in addition to speaker ids) or modeling multimodal conversations (images, actions, etc.), though this may require more capable language models. While we demonstrated the promise of this method using interactive conversations, it can be applied to turn language models into interactive models for any kind of event sequence, i.e., sparse-over-time world models. We hope that this method will facilitate more flexible interaction with the underlying capabilities of language models and enable new applications in fields such as gaming and entertainment.

## ETHICAL CONSIDERATIONS

While work improving the ability to simulate real-time interactive conversations can make language models more useful or delightful, it also poses risks for fraud and manipulation. In order to mitigate these risks, we limit our work to simulating natural conversations in text, a medium which is perceived as less trustworthy than audio or video. (While we simulate the timing aspects of spoken conversation, our generations are still easily distinguished from real speech.) We provide only proofs of concept with small datasets, and do not scale up to sizes where these capabilities would become more refined. We also do not study goal-directed methods which could be used to steer a model to execute fraud.

We believe that it is valuable to expose this capability overhang so that the community can respond with appropriate measures. For example, a better understanding of the amount of data needed to impersonate someone with a generative clone could affect how much conversational data users are comfortable sharing publicly on social media, or motivate end-to-end encryption/disappearing messages to prevent private data leakage in the event of hacking. Developing interfaces for language models that are not immediately distinguishable from humans could also help to evaluate extreme risks like deception and persuasion in frontier models (Shevlane et al., 2023), to the extent that people react differently to communication that they perceive to be from a model vs. another person. Bad actors are already capable of sophisticated deepfake scams and aren't exactly forthcoming about their methods.

There are also ethical considerations when simulating real people or fictional characters absent ill intent, such as privacy and the effects of parasocial relationships; these tend to be general concerns that are not strictly related to real-time interactivity. See Morris & Brubaker (2024) for an in-depth discussion of these factors. In terms of the specific datasets we used in this paper: We used our own instant messenger history with the consent and active involvement of both participants, and do not release the data/model for privacy reasons. The U.S. Supreme Court's oral arguments are inherently public and the conversation is in a specialized legal domain rather than anything that would encourage parasocial relationships. We model only text transcripts and use generic text to speech voices (i.e., we do not contribute methods to impersonate any of the speakers).

## REPRODUCIBILITY STATEMENT

We publicly release the code for our case studies at this link. We do not release our own personal instant messenger history for reasons of privacy, but you can reproduce the instant messenger case study by bringing your own data. The data for the spoken conversation case study is public and can be reproduced.

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

# A Causal Rejection Sampling Algorithm with Speculative Decoding

In Algorithm 2, we present a description of causal rejection sampling including speculative decoding. For simplicity, we describe the speculative rejection sampling as if it rejects or accepts an entire event, but in implementations where events are composed of multiple tokens, the acceptance/rejection acts in finer granularity on tokens (so a prefix in a speculated event can be accepted, and only the rest has to be resampled).

Note that in order to maintain the validity of the rejection sampling, we must condition the draft distribution on $t_i \geq T$, because if $\hat{t}_i$ had been $< T$ it would have already been finalized and we would not be considering it for rejection sampling. Renormalizing this correctly in timestamps consisting of multiple tokens requires some finesse. For our instant messenger case, this is further complicated by the fact that as interruptions come in, the same timestamp in a draft message may change format. For example, after a message planned for `:02;17.8`, a generation may plan for `:03;52.0`, but after an interruption at `:03;24.7`, the draft must be reinterpreted to have a timestamp of `;52.0)`. We expect that speculation is not worth the implementation burden unless you are using a simple custom vocabulary for timestamps or pressing up against performance limits. In our experiments performance was adequate without it.

---

**Algorithm 2** Causal rejection sampling with speculation

---

$e \leftarrow []$      ▷ event history
$i \leftarrow 0$      ▷ current event index
$c \leftarrow (\varnothing, \varnothing, \varnothing)$      ▷ candidate for the next message
$r \leftarrow$ false      ▷ whether a candidate was just rejected
**while** true
    $i \leftarrow i + 1$
    **try**
        $(\hat{t}, \hat{s}, \hat{m}) \leftarrow c$
        **if** $\hat{t}$ is $\varnothing$
            $P \overset{\triangle}{=} p(e_i | e_1, ..., e_{i-1})$
            **if** $r$
                $Q \overset{\triangle}{=} p(e_{i-1} | e_1, ..., e_{i-2}, t_{i-1} \geq T)$
                $c \leftarrow (\hat{t}, \hat{s}, \hat{m}) \sim norm(max(0, P - Q))$
                $r \leftarrow$ false
            **else**
                $c \leftarrow (\hat{t}, \hat{s}, \hat{m}) \sim P$
        **wait** until $\hat{t}$
        $t_{cur} \leftarrow \hat{t}$
        **if** $\hat{s}$ is $S$
            $c \leftarrow (\varnothing, \varnothing, \varnothing)$
            $i \leftarrow i - 1$
            **continue**
    **catch** user input $(T, S, M)$
        $e_i \leftarrow (T, S, M)$
        $t_{cur} \leftarrow T$
        $(\hat{t}, \hat{s}, \hat{m}) \leftarrow c$
        **if** $\hat{t} + t_{react} < T$
            **if** $\hat{s} \neq S$
                $P \overset{\triangle}{=} p(e_{i+1} = c | e_1, ..., e_i = (T, S, M))$
                $Q \overset{\triangle}{=} p(e_i = c | e_1, ..., e_{i-1}, t_i \geq T)$
                **if** $Q \leq P$
                    **continue**
                **else**
                    **if** $u \sim U[0, 1] > 1 - \frac{P}{Q}$
                        **continue**
                    **else**
                        $r \leftarrow$ true
            $c \leftarrow (\varnothing, \varnothing, \varnothing)$
        **continue**
    $e_i \leftarrow c$
    **emit** $c$
    $c \leftarrow (\varnothing, \varnothing, \varnothing)$

---

## B    CAUSAL REJECTION SAMPLING ALGORITHM WITH RETCONNING (WITHOUT SPECULATIVE DECODING)

In Algorithm 3, we present a description of causal rejection sampling with retconning, which supports streaming word-level ASR input where previous inputs may change retroactively given new context. For simplicity, we do not include speculative decoding, though it could also be applied upon retconning.

---

**Algorithm 3** Causal rejection sampling with retconning (without speculative decoding)

---

$i \leftarrow 0$                ▷ current event index
$e \leftarrow []$                ▷ event history
$c \leftarrow (\varnothing, \varnothing, \varnothing)$      ▷ candidate for the next message
**while** true
    $i \leftarrow i + 1$
    **try**
        $(\hat{t}, \hat{s}, \hat{m}) \leftarrow c$
        **if** $\hat{t}$ is $\varnothing$
            $c \leftarrow (\hat{t}, \hat{s}, \hat{m}) \sim p(e_i | e_1, ..., e_{i-1})$
        **wait** until $\hat{t}$
        $t_{cur} \leftarrow \hat{t}$
        **if** $\hat{s}$ is $S$
            $c \leftarrow (\varnothing, \varnothing, \varnothing)$
            $i \leftarrow i - 1$
            **continue**
    **catch** user input $(T, S, M)$
        $e_i \leftarrow (T, S, M)$
        $t_{cur} \leftarrow T$
        $(\hat{t}, \hat{s}, \hat{m}) \leftarrow c$
        **if** $\hat{s} = S$ or $\hat{t} + t_{react} < T$
            $c \leftarrow (\varnothing, \varnothing, \varnothing)$
        **continue**
    **catch** user retcon $j, (T, S, M), t_{cur}$
        $e_j \leftarrow (T, S, M)$
        $(\hat{t}, \hat{s}, \hat{m}) \leftarrow c$
        **if** $\hat{s} = S$ or $\hat{t} + t_{react} < t_{current}$
            $c \leftarrow (\varnothing, \varnothing, \varnothing)$
        $i \leftarrow i - 1$
        **continue**
    $e_i \leftarrow c$
    **emit** $c$
    $c \leftarrow (\varnothing, \varnothing, \varnothing)$

---

## C  FINETUNING DETAILS

We finetune Pythia 160M, Pythia 1.2B, Pythia 12B, Gemma 2B, and Llama 2 7B. We finetune for
several epochs with learning rate $10^{-5}$ and batch size 512. We use early stopping on validation loss
(computed once per epoch) with a minimum delta of 0.01 and a patience of 3.

## D  IN-CONTEXT LEARNING DETAILS

For our ICL experiments using GPT and Claude, we use the following system prompts and decode
with default sampling parameters.

**Instant messenger dialogues:**

```
Your job is to continue instant messenger conversations between two individuals
inspired by a partial transcript of their chat history. Your generated
conversations must be new (i.e., they should not appear in whole or in part in
the transcript), but they should be stylistically and factually consistent with
the transcript. You must preserve the characterization of both individuals as much
as possible. DO NOT include anything in your response except a continuation of the
provided conversation transcript in the same format as the chat transcript. Output
nothing else, either before or after the continuation.

Each message in the chat transcript is formatted as follows:

<timestamp><user><message><delimiter>

A full transcript consists of many messages in this format concatenated
together without any whitespace. A sample message is given below:

2023June04Su+01:53;42.7Anow that you mention it-------+

The <timestamp> field includes the year (e.g. "2023"), the month (e.g. "June")
the date (e.g. "04"), one or two letters denoting the weekday (e.g. "Su") , and
then the time in UTC (e.g. "01:53;42.7"), all concatenated without spaces in
that order. If any part of the timestamp is the same as in the previous message,
it is omitted to save space.

The <user> field is either "A" or "B".

The <message> field is an arbitrary string (in this case "now that you mention
it").

<delimiter> is always "-------+".
```

**Spoken conversation:**

```
Your job is to continue timed speech transcripts of Supreme Court arguments.
Your generated transcript completions must be new (i.e., they should not appear
in whole or in part in the transcript), but they should be stylistically and
factually consistent with the transcript. You must preserve the characterization
of the speakers as much as possible. DO NOT include anything in your response
except a continuation of the provided transcript in the same format as the
transcript. Output nothing else, either before or after the continuation.

Each word of the transcript is formatted as follows:

<timestamp><speaker><word>

A full transcript consists of many words in this format concatenated together
```

```
with one space in between. A sample message is given below:

131Gwe'll
```

The `<timestamp>` field is three digits denoting the seconds place of the time, the decisecond, and the centisecond.

The `<speaker>` field is a single capital letter.

The `<word>` field is a single word.

```
<transcript>
```

## E    EXPERIMENTAL DETAILS

For the instant messenger human ratings: offline ratings are averaged over 20 examples per model and online ratings are averaged over 5. For the spoken conversations human ratings: offline ratings are averaged over 10 examples per model and online ratings are averaged over 5.

## F    QUALITATIVE EXAMPLES

See below for examples of ground truth and generated conversations, for both the instant messenger dialogue and spoken conversation case studies.

See Figure 5 for a ground truth example, Figure 6 for a Pythia 160M example, Figure 7 for a Pythia 1.4B example, Figure 8 for a Pythia 12B example, Figure 9 for a Gemma 2B example, Figure 10 for a Llama 2 7B example, Figure 11 for a Claude 3 Sonnet example, and Figure 12 for a GPT-4 Turbo example.

See Figure 13 for a ground truth example, Figure 14 for a Pythia 160M example, Figure 15 for a Pythia 1.4B example, Figure 16 for a Pythia 12B example, Figure 17 for a Gemma 2B example, Figure 18 for a Llama 2 7B example, Figure 19 for a Claude 3 Sonnet example, and Figure 20 for a GPT-4 Turbo example.

```
:55;50.0AI think chatbot + speechbot could be a nice paper together
:56;16.6Athe contribution is real-time language modeling with
         timestamp decoding

                              ;19.1Bthe worst thing about this paper
                 ;25.0Bis that some combination of the paper itself
                                  ;28.5Band the cluster difficulties
   ;38.4Bmakes it feel like I'm just endlessly repeating the same
                                                             task

;43.8Ayes
```

Figure 5: **Ground truth instant messenger example.**

```
:11;35.5Ahttps://www.[redacted]/watch?v=jG-IHVtU-Y

                              :48;02.8Bhttps://[redacted]/C4vNE7CdK0

;05.5Ahttps://[redacted]/2/?l=ms/4401127580322316

                                        :55;18.7Bsad

;29.6Ahttps://www.[redacted]/videos/202816392889398/
;34.0Amy comment is so rikt'd
:56;01.5Ahttps://[redacted]/status/1716488523789759696
;05.1Aalso those responses
;08.8Alol
;15.8Athe adults are not actually that bad
;32.3AI mean
;32.8Athey sent humans attached
;35.2Ain this case
;37.8Athe two houses are just some random images of babies tomoe
:58;06.2Awould look like a tree in hans graham's apple courtyard
:59;23.4Athey're just natural trees
;38.3Aare they scientific creation?
+22:10;08.4Athe place where black hole black holes go from is not
              really an underground world
;37.9Aand they're everywhere
:12;35.4Ayes

                              ;40.8Bthey're moving fingers
```

Figure 6: **Pythia 160M instant messenger example.**

```
                                                          ;07.5Blol
                                                     ;17.6Bso good
                               ;22.3B"if all this other stuff were true"
                      ;27.0B"you probably should not have lied to the world"
                    :09;24.7BI need to stop being so cynical about this
                                                 ;26.5Bit's so good
              ;40.0BI'll only be cynical if I have to justify it somehow

      :14;33.8Ayes
      ;53.0AI can't imagine why you would have that
              in your head though

                                        :15;07.9Bit's pretty funny
                            ;11.4BI told you how my friend who is like
                                             ;13.81.4BM students
                                                    ;16.5Bgets in
                                ;19.0Band asks me to come to his
                                          ;22.1BI am a sophomore
                                               ;24.2Bit's like
                                        ;26.9Bit's the same age
                            ;29.9Bbut he's in a more selective school

      ;47.5Ayes I don't understand how people are this cynical

                          ;53.8Bhow did you become a social critic so late
                          :16;21.4Band have this incredible network
                                        ;23.9BI am in that category too
             ;31.7Bthere's probably a social media layer under the network
               ;36.8Bwhere people who are in the school in a certain way
                                        ;39.3Bare probably very good at it
             ;44.7Blike maybe a few years ago when I first found my niche
                                  I probably had to work really hard
                                            ;46.6Bto get attention
                          ;56.0Bbut you kind of need to do it constantly

      .5Ayes that's what I was thinking of when I said the other day

                                              ;57.4Bit's like
```

Figure 7: **Pythia 1.4B instant messenger example.**

```
;44.0Ait's going to talk about earth
:48;17.4Alol
;18.6Ait did
:52;40.7AI can't believe how good it is
:53;05.8Ait can apparently hear a person's breathing and then say
          what they think is the most likely reason for that
;10.1Alike
;18.7A"I'm guessing it's the result of you exhaling"
;19.4A"lol"
;22.4AI didn't even understand it

     :57;00.1Bis it so good at understanding what is being asked
   :58;05.8Bit can tell what people are talking about pretty well

;26.1Athat is impressive
;33.3Abut what do they mean by that

            ;34.8Bit can anticipate what people will ask it about
                                  ;39.6Band guess correctly

;44.7Athat's kind of an illusion I think
;51.0Aespecially at test time

                                    :59;02.8Bit's not an illusion
                                         ;04.2Bit's a prediction
                                ;17.2Bthis one is from the book
                             ;26.0Bit's just really fast and deep
                 ;36.8Bmaybe it's really good at interpreting images
       ;45.3BI think it has some kind of image memory of whatever
                                   it saw the person doing previously

:59;24.9Ayes but it was probably trained with that data at least
;29.4Abut maybe they are good at understanding like
;32.4A"which object in the picture"
;33.9Arather than

                             .9Bit probably doesn't remember like
                                  ;38.5B"I've seen the sun before"
   ;44.3B"let me predict that you'll say 'it looks bright on the
                                              left side"'"
                                             ;45.3Byes it is
```

Figure 8: **Pythia 12B instant messenger example.**

```
:19;01.7Ait's hard to say

                                       ;08.8Bnot as long as you do stuff
                                         ;11.9Byou can still do lots

       ;16.4Ayes

                                         ;21.0BI get exhausted though
                                ;35.3BI'm so tired after a day like that

       ;44.1Alol

                             ;58.7Bso you could definitely do more work

       :20;03.0AI get more motivated to start
       ;09.1Abut then I'm constantly worried

                                         ;14.0Bthat's probably a positive!

       ;17.9AI'll get bored

          ;25.4Bmaybe less productive than someone without that feeling

       ;28.0Aso I have to just do something

                                     ;30.5Bif you need to have things done

       ;34.3Ayes

                                     ;35.2Bthen have more things to do

       ;37.5Athen why not work

                                ;42.4BI literally can't work on anything
                                                       ;43.6Bfor days

       .9Aif you're not already doing work
       ;45.8Ayou need to stop thinking
       ;51.5Aand just do it

                             :21;24.1Blike this week has been unproductive
```

Figure 9: **Gemma 2B instant messenger example.**

```
                                                  ;41.8Bit's obviously false
                                     ;47.3Bthe fact that it's true is the proof
                                                        ;56.0Bit's so good
                                                             ;58.1Bso deep
                                                       :50;01.9Bthe irony
                                                ;25.2Bbut then it's like
                        ;32.3Bwhat are the implications of the obvious truth
                ;58.7Bwho is going to write a textbook called "complexity and
                                                               universality"
             :51;17.6Bthere's no way any human in this class would get it
                           ;46.6BI guess it's just something professors say
                        :52;07.1Bjust one page in every 300 page textbook
                                    ;15.4Band like three times in the rest
                                      ;18.9Bwhen they talk about complexity

   :53;00.5Amy friend sent me a list of all her classes so far
   ;04.4Aand they're all so interesting
   ;09.7Alike one was a class on china
   ;13.4Aand another was the music of west africa
   ;22.2Athat reminds me
   ;25.8Awhat happened to your internships

                                       ;33.0BI've just been sitting around
                                      ;40.2Bwaiting for interviews to happen
                          :54;02.2BI got a call about a different one today

   ;03.4Af

                                    ;09.6Bbut I don't want to be doing that

   ;13.0Ayes

                          ;17.4BI'm waiting for the ML interviews to happen
                      ;25.2Bhopefully the university one will come through
                             ;29.1Bsince that's the most promising one

   ;56.9Awhat else were the other two
```

Figure 10: **Llama 2 7B instant messenger example.**

```
  :41;02.7Bj'm surprised my senses aren't triggering too quickly
 ;24.4Bit feels like you'll have to spend half an hour breathing

:43;00.5Alol I see the pop quiz early tomorrow

                 :44;34.2Bso I can only focus on writing the essay

;37.7Athen I'll meet you at 12:45
;41.2Aprobably I'll meet you anyway
;43.4Aand I have an appetite for reading your short stories
     one day

                                              :46;05.3Bhm

:47;58.4AI just finished this letter

 :49;02.4Bthat's just the end of today. I just want to sit still

;46.0Aand read it

                                    ;47.2BI have an expectation that
                              ;49.3Bit's going to be fun to write it
                              ;52.9Bbut I don't want to write it
                                               :50;11.0Bsmh
                              :51;49.1Bit's supposed to be rpg

.8Abut whoops
;52.0AI'm so anxious to finish it

;53.6Bbut then I'll just need to study about word choice at least
```

Figure 11: **Claude 3 Sonnet instant messenger example.**

```
;22;33.5APractically inviting the memes at this point

            ;23;45.9Bcan't believe the world we live in sometimes

;27;14.2Aright?
;29;00.7Aevery day feels like a satirical novel

                              ;31;52.3Bexcept it's painfully real

;35;06.8Aexactly
;38;29.5Aanyway did you finish that analysis?

                              ;40;03.1Byeah, nearly there
                    ;42;15.6Bgot sidetracked by the news tbh

;45;04.9Aunderstandable
;47;11.3Asend it over when you're done?

                                    ;49;22.8Bwill do
                    ;51;30.9Bneed a distraction from this madness

;53;45.2Asame here
;55;57.9Alet's grab coffee later, need to vent
```

Figure 12: **GPT-4 Turbo instant messenger example.**

```
229Aany
259Aaction
293Aunder
395Acould
411Abe
425Abrought
457Aincluding
497Aa
505Aquitam
561Aaction
653Aor
683Aincluding
723Aa
729Aretaliation
815Aaction
907Abut
935Athat
961Areading
001Ais
017Anot
043Aessential
105Ato
214Athe
226Aposition
284Athat
298Awe
316Aadvocate
368Ahere
                                    422Cso
                                  442Cthat
                                532Ccould
                              580Cexclude
                                 746Ceven
                                    768Ca
                          776Cretaliation
                              882Caction
                                  996Cby
                                  020Can
                          032Cindividual
                                 122Cwho
                              199Cdoesn't
                                241Cknow
                                291Cwhen
                                 323Cthe
                                427Cwhen
                                 449Cthe
                          469Cindividual
                                 631Cthe
                             643Cofficial
                                  685Cof
                                 693Cthe
                              707Cunited
                              747Cstates
                                837Cfound
                                 867Cout
                               881Cabout
                               907Cthose
                                935Cfacts
```

Figure 13: **Ground truth spoken conversation example.**

```
751Awhite
799Athat
821Ato
837Ayour
853Aoffice
903Ais
999Avery
949Alikely
057Aunder
079Athe
095Aoriginal
155Achapter
209Aso
211Aif
251Ayour
263Amajority
321Acompels
443Athe
456Aattaches
495Athe
511Aperson
553Ais
587Ato
627Amurder
999Aas
039Aby
049Athe
063Aattaches
127Ais
141Athe
171Awhite
222Aman
247Ato
271Ayou
285Aand
299Amays
327Ato
361Ame
375Athe
407Aperson
435Aoffering
475Ainformation
559Afor
565Awork
625Awas
665Athe
681Atype
697Aof
707Aan
719Aprogressive
853Aexpression
963Aof
981Aspeed
989Ashifts
061Ado
081Athat
107Aoccur
```

Figure 14: **Pythia 160M spoken conversation example.**

```
719Ayork
745Alaw
773Asay
805Ayou
827Acan't
851Ado
875Aanything
975Ai
983Asuppose
031Ayou
059Aknow
085Ahow
099Ato
109Ado
121Asomething
165Aover
180Athere
                                   223Ji
                                   229Jdon't
                                                                      239Gknow
                                                                     255Gthat's
                                                                     279Gwhere
                                                                       299Gi'm
                                                                    319Ggetting
                                                                     347Gfrom
                                                                     365Gyour
                                                                    391Ganswer
                                                                       435Gthe
                                                                    449Gquestion
                                                                        487Gis
                                                                       501Gwhy
                                                                     535Gisn't
                                                                     567Gthat
                                                                     585Ggoing
                                                                       611Gto
                                                                       623Gbe
                                                                     637Gdone
                                   829Jwell
                                   851Jthat's
                                   873Jtrue
                                   887Jbut
                                   933Jone
                                   945Jway
                                   963Jto
                                   977Jdo
                                   001Jit
                                   019Jis
                                   033Jto
                                   049Jget
                                   073Jthe
                                   089Jnotice
                                   127Jto
                                   139Jthe
                                   155Jbank
```

Figure 15: **Pythia 1.4B spoken conversation example.**

```
312Aflames
407Agoing
469Athere
488Awasn't
515Aa
517Afire
550Ajust
566Agoing
595Athat
613Away
650Athen
670Ait's
684Aan
694Ainvalid
740Asearch
770Aright
                                            788Fthat
                                            800Fwould
                                            816Fbe
                                            832Fan
                                            842Finvalid
                                            870Fsearch
                                            918Fif
                                            932Fthere
                                            948Fwas
                                            964Fnever
                                            988Fa
                                            996Ffire
                                            036Fat
                                            042Fthe
                                            050Fhouse
                                            076Fif
                                            084Fthere's
                                            108Fnever
                                            140Fa
                                            148Ffire
                                            170Fat
                                            180Fthe
                                            190Fhouse
                                            224Fnow
                                            240Fjustice
                                            270Fstevens
                                            300Fasked
                                            316Fthe
                                            326Fquestion
                                            360Fthe
                                            374Ffacts
                                            406Fare
                                            458Fwell
                                            478Fmay
                                            498Fnot
                                            522Fhave
                                            540Fbeen
                                            560Fa
                                            566Ffire
                                            584Fin
                                            592Fthe
                                            604Fhouse
```

Figure 16: **Pythia 12B spoken conversation example.**

```
372Abut
400Ahow
430Ado
444Awe
460Aknow
482Awhen
498Athere's
524Aa
532Asecond
568Astep
606Arequired
654Ai
668Athink
692Athat
708Athe
720Areason
758Athat
774Awisconsin
834Afailed
864Ato
882Ado
902Ait
920Ais
932Abecause
962Aof
976Ajustice
016Ascalia's
072Aconcern
134Aabout
156Athe
168Aburdens
218Aon
230Adue
250Aprocess
                                                    352Bwell
                                                   366Bthat's
                                                      390Bin
                                                    402Bpart
                                                       426Ba
                                                  434Bproblem
                                                     460Bwe
                                                    472Bhave
                                                    504Bwith
                                                     516Bthe
                                                530Brationally
                                                  576Brelated
                                                    614Btest
                                                   644Bwhich
                                                      666Bis
                                                    684Bthat
                                                      762Bif
                                                       782Bi
                                                     798Bput
                                                       820Ba
                                              830Bhypothetical
                                                     900Bon
                                                    908Bthe
                                                   918Btable
```

Figure 17: **Gemma 2B spoken conversation example.**

```
096Afor
110Aexample
170Athat
188Acame
212Ainto
232Aevidence
274Athat
304Ahe
318Ahad
338Aa
346Alot
362Aof
370Adifferent
398Adrivers
442Alicenses
520Athere
538Awas
558Aevidence
590Afrom
608Athe
618Amaricopa
674Acounty
714Aattorney's
764Aoffice
790Athat
804Ahe
814Ahad
844Anine
876Aprevious
934Afelony
976Aconvictions
050Ai'm
062Asorry
096Aeight
112Aprevious
160Afelony
198Aconvictions
297Athere
312Awas
339Aevidence
373Athat
397Ahe
415Ahad
481Ahad
497Aan
507Aextensive
561Afraud
595Abankruptcy
655Acase
709Abefore
```

Figure 18: **Llama 2 7B spoken conversation example.**

```
131Awe'll
563Acounty
676Asoil
                    718Gand
                    732Gwater
                                766Hconservation
                                846Hdistrict
                                972Hversus
                                013Hthe
                                027Hunited
                                071Hstates
                                007Hmr
                                021Hbrowning
                                                    217Bmr
                                                    245Bchief
                                                    267Bjustice
                                                    331Band
                                                    343Bmay
                                                    359Bit
                                                    369Bplease
                                                    405Bthe
                                                    417Bcourt
                                                    503Bthe
                                                    527Bissue
                                                    565Bin
                                                    575Bthis
                                                    603Bcase
                                                    697Bis
                                                    713Bwhether
                                                    747Bcongress
                                                    831Bexpressly
                                                    901Bprovided
                                                    959Bfor
                                                    977Ba
                                                    985Blimitations
                                                    061Bperiod
                                                    103Bfor
                                                    125Bretaliatory
                                                    199Bdischarge
                                                    269Baction
                                                    325Bunder
                                                    343Bthe
                                                    353Bfederal
                                                    385Bfalse
                                                    421Bclaims
                                                    469Bact
                                                    551Bthe
                                                    567Bsix-year
                                                    613Blimitation
                                                    679Bperiod
                                                    764Bset
                                                    792Bout
                                                    873Bin
                                                    897Bsection
                                                    002B3731b
```

Figure 19: **Claude 3 Sonnet spoken conversation example.**

```
464Agive
480Aa
492Awritten
508Astatement
552Awithout
570Athe
582Apresence
609Aof
617Aan
625Aattorney.
                                                            682CThat's
                                                            700Cclear,
                                                              716Cbut
                                                              728Cthe
                                                              736Cfact
                                                              744Cthat
                                                               752Che
                                                               756Cwas
                                                            760Cwilling
                                                                776Cto
                                                             780Cspeak
                                                            794Corally
                                                            823Cwithout
                                                              841Cone
                                                            857Cdoesn't
                                                         865Cnecessarily
                                                             925Cmean
                                                               945Che
                                                          957Cunderstood
                                                              979Cthe
                                                         003Cimplications
                                                             055Cfully.
                          102GThat
                          118Gis,
                          132Gdid
                          146Ghe
                          160Gunderstand
                          188Gthat
                          200Gan
                          212Goral
                          226Gstatement
                          262Gcould
                          274Gstill
                          286Gbe
                          298Gused
                          314Gagainst
                          330Ghim
                          344Gin
                          352Ga
                          364Gcourt
                          376Gof
                          384Glaw
                          408Gjust
                          420Gas
                          432Geffectively
                          472Gas
                          486Ga
                          494Gwritten
                          516Gone?
```

Figure 20: **GPT-4 Turbo spoken conversation example.**

