# OpenReview forum: "Modeling Real-Time Interactive Conversations as Timed Diarized Transcripts"
_ICLR.cc/2025/Conference — Submitted to ICLR 2025_

### Official Review · Reviewer_JDak · 2024-11-03

**Soundness:** 2
**Presentation:** 2
**Contribution:** 2
**Rating:** 3
**Confidence:** 2

**Summary:**

This paper presents a novel approach for achieving real-time response in dialogue systems by modeling timed, diarized transcripts and decoding through causal rejection sampling. The model is tested on both instant messenger dialogues and spoken conversations, with results demonstrating its ability to sustain real-time interactivity.

**Strengths:**

The proposed dialogue system supports multi-speaker, simultaneous conversations, marking a significant advancement over traditional turn-by-turn dialogue models. Additionally, the method’s design allows it to be easily integrated into various models, demonstrating its versatility and broad applicability.

**Weaknesses:**

1. Additional Evaluation Metrics to Consider
- While the model demonstrates high generation bandwidth, how does your approach ensure the model engages at the right time rather than speaking continuously? In other words, how does your method balance high generation rates with appropriate turn-taking?
- An ablation study comparing models without rejection sampling and those fine-tuned for next-turn speaker and response prediction could help validate the model design.
2. Positioning and Contribution

This paper proposes a system design for real-time conversation. However, its novelty from a learning and evaluation perspective isn’t fully highlighted. I’d be open to discussing potential ways to position this paper’s contributions for ICLR.

**Questions:**

NA

---

> ### Author Response · Authors · 2024-11-23
>
> We are grateful that the reviewer considers our work a “significant advancement” and that our method is versatile and broadly applicable. We respond to their specific points below:
>
> **Re message timing**
>
> The high generation rates and appropriate turn-taking are orthogonal. The capability for high generation rate is a function of our algorithms/their implementation (modulo the small assumptions about human reaction time), while the turn-taking and content of the conversation is learned by the weights of the model from the training data. Better language modeling of transcripts (where continuous speech of the kind you describe is not observed) corresponds to better turn-taking. Empirically, we find that continuous speech of this kind does not occur in model generations: in Figure 4, we provide quantitative data showing that the timings of generated transcripts are broadly realistic, and in the appendix, we provide samples from our various trained models, where users do in fact take realistic turns.
>
> We argue that a key strength of our method is specifically that it does *not* involve any sort of manual intervention in message timing, instead allowing the LLM to model message timings and content jointly in subtle ways that would be impractical with a hand-engineered solution.
>
> **Re. ablation**
>
> We’re not sure we understand exactly what the reviewer is requesting. Rejection sampling happens exclusively at inference time, not during training, and is simply a way to handle interruptions from a real-time user. A model that does not use causal rejection sampling would not be a real-time interactive model (it would ignore intervening user input). Our quantitative experiments in Figure 4 as well as the generations in the appendix are all simply sampled from the model without rejection sampling, whereas our interactive human evaluations included it.
>
> **Re. novelty**
>
> We’d be happy to further clarify the novelty of our work.
>
> We offer a new method to use pretrained LLMs to take timed dialogues, or more broadly any event series that could be converted into a timed transcript, and turn them into real-time interactive simulators. There was previously no standard method for generative and interactive modeling of this kind of data, and our method offers many benefits (leveraging LLM pretraining, sustaining high interactive bandwidth, supporting long context windows efficiently, simplicity) over idiosyncratic methods from works like CICERO that coordinated many special-purpose models or use heuristics rather than end-to-end modeling. These advantages are highlighted throughout the paper. The novelty therefore comes in task framing and modeling, and essentially our new method brings this class of modeling tasks into the LLM era (enabling many new applications). We do not claim evaluation novelty as a main contribution (though as we describe on line 321, our human fidelity rating task is an interesting new long context LLM eval).
>
> We hope we have addressed the reviewer’s concerns. If there remains anything standing in the way of an endorsement of the paper, please let us know.

---

### Official Review · Reviewer_63e3 · 2024-11-04

**Soundness:** 2
**Presentation:** 2
**Contribution:** 2
**Rating:** 5
**Confidence:** 4

**Summary:**

The paper introduces a method for simulating real-time interactive conversations using pretrained text-only language models, incorporating two key modifications. First, it employs timed diarized transcripts to represent each timestamped event, with each entry consisting of a timestamp, speaker ID, and message content. The model is tasked with predicting the probability of the next event based on event history, and sampling can proceed token by token, similar to standard causal language model text generation. During inference, a technique termed causal rejection sampling enables real-time interaction by discarding and resampling responses when interrupted by the user, thus adapting to dynamic input. To improve response speed during user interruptions, the authors also introduce two enhancements: (1) accounting for model generation latency and user reaction time and (2) applying a modified speculative decoding method to reuse partially generated responses before interruption.

The method is evaluated in two domains: instant messaging and spoken conversation. For instant messaging, the authors processed a 9-year message history between the first authors, resulting in 37 million characters in the diarized transcript format. For spoken conversations, they utilized a 1000-hour subset of U.S. Supreme Court oral arguments, totaling 33 million characters. This spoken data was converted into word-level transcripts with precise timing using an ASR engine. Experiments involved both open-source LLMs (Pythia, Gemma, LLaMA) ranging from 160M to 7B parameters, and state-of-the-art proprietary LLMs. Evaluation metrics included document-level perplexity, offline human evaluations ranking generated continuations, and online human evaluations involving direct interaction with the model. Key findings indicate that (1) the method meets real-time interactivity constraints on feasible hardware (e.g., a 40GB A100 GPU for a 7B LLaMA 2 model), and (2) larger, high-quality LLMs generally perform better in real-time interactive conversation modeling.

**Strengths:**

- Adapting text-only LLMs to model real-time interactive conversations is an important yet underexplored problem with the potential to unlock new applications without the need for costly retraining. The proposed method is straightforward but effective, demonstrated across a variety of LLMs from different model families and scales, and applicable to both fine-tuning and in-context learning.
- The paper includes both offline and online human evaluations in addition to automatic metrics, which are particularly valuable for assessing the performance of LLMs.

**Weaknesses:**

- A critical aspect of real-time conversation is that the user can interrupt at any moment, requiring the model to discard its current response and handle the new input immediately. However, while the proposed method has the potential to address this, it is not thoroughly tested in the experiments. There is not even a qualitative example showing such behavior from the trained models.
- Using instant messaging dialogues to evaluate real-time interactions may not be ideal, as people’s response habits vary depending on availability, typing speed, and other factors. Modeling response timing directly may be less meaningful in this setting, and the turn-based nature of messaging lacks the fluidity and interruption dynamics of real-time conversations.
- The human evaluation lacks details, such as the number of conversations evaluated per method, the evaluation rubric, and consistency across ratings. It is unclear how the conversations are compared or if a consistent standard was applied, which is particularly important given the challenge of evaluating long texts.
- The control tokens introduce significant overhead in the spoken conversation domain without an optimized tokenizer that treats numbers from 0 to 999 as single tokens. Since existing models may not support such tokens, this limits the method’s practicality when used with standard tokenizers.

**Questions:**

Q1: Could you clarify what is meant by the "first 95%"? Is this based on chronological order? If so, wouldn’t this temporal split risk introducing shifts in topics or conversational styles over time, potentially affecting the robustness and validity of the evaluation?
> We use the first 95% of the messages as the train set, the next 2.5% as a validation set, and the last 2.5% as a test set.

Q2: In Figure 4, timing is evaluated based on the delays between successive messages. Could you clarify why only delays are measured rather than both delays and any potential early responses? Additionally, how is the alignment between generated and ground-truth messages determined? If the generated messages differ significantly in content from the ground-truth, what is the meaning or value of comparing the timing between two sets of messages that may not correspond closely in their content?

Q3: Why were different evaluation metrics used for human assessments in the instant messaging and spoken conversation settings? Could you explain the reasoning behind this choice and how it aligns with the specific goals of each domain?

---

> ### Author Response · Authors · 2024-11-23
>
> We thank the reviewer for their detailed reading and are glad they found our method straightforward, effective, and broadly applicable. We address their concerns below:
>
> **Re. interruptions**
>
> This is a fair point, but it gets at some of the difficulties of evaluating the content of a generated transcript. This was one of the things we focused on during the human evaluations—at some point in the conversation, the rater would often send an outlandish nonsequitur to the language model and gauge how realistic its response was—and it did factor into both consistency and fidelity scores. The sample generations do not currently include interruptions per se (we tried not to cherry-pick, filtering only for privacy and anonymity), but there are instances of fast-paced conversation—towards the bottom of Figure 10, for example, the model “sends” a message to itself one second after the previous message, and the other “user” responds appropriately. Figure 4 does not evaluate content, just timing (see the next section of this rebuttal), but it also shows that the model can maintain a realistic message cadence even during fast-paced conversation.
>
> We’d argue that, while interruptions are an important ingredient of real-time interactivity, other, more elementary ones—like topical coherence, realistic message timings, fidelity to speaker identities, self-repair—are just as important. We are not aware of a competing method that handles these in a general way as well as ours (let alone interruptions!).
>
> Finally, we’d like to point out that, in a sense, *any* message the user sends is an interruption: before the language model receives a message from the user, it had some plan for a future message that it now needs to change in some way.
>
> **Re. instant messenger dialogues**
>
> While it’s true that instant messenger chats lack some of the fluidity of real-life conversations, a) we’d argue that modeling them faithfully is still highly nontrivial, and modeling this unique form of data is beyond the existing state of the art, b) instant messenger data is much more readily available in the correct format (anyone can apply our method to their own chat histories), and c) it does still feature significant fluidity—looking at the ground-truth distribution of message timings in Figure 4, we see that a significant fraction of messages are less than five seconds apart. A large number (hundreds of thousands of messages) are less than one second apart. Not all of these represent interruptions, as we do not distinguish between users here, but many do. We were able to chat with the model in real time, and once the system engages in a conversation it's unlikely to "step away" without warning
>
> **Re. evaluation details**
>
> We will include more detailed metadata in the final version of the manuscript (expanding Appendix E).
>
> **Re. control token overhead**
>
> While it’s true that the control tokens introduce overhead for un-optimized tokenizers, we argue that this overhead is not “significant” in the sense of being prohibitive. The bottom left pane of Figure 3 shows that while an optimized tokenizer like you describe does indeed lead to lower generation speed requirements, Llama’s real tokenizer is not very far behind, and its rates still fall well within the range that can be achieved on commodity hardware. Alternatively, you could always introduce additional tokens with finetuning. Widely used tokenizers have many “spare” tokens set aside for similar purposes.
>
> **Re. Q1**
>
> Yes, this is based on chronological order, and intentionally so. Holding out a test set across makes the evaluation even more robust/valid, because we test the kind of distribution shift over time that is experienced in deployment (when training on historical data).
>
>
> **Re. Q2**
>
> Models trained in this fashion can be evaluated along two primary axes: content and timing. It is relatively hard to quantitatively evaluate the content of messages; for that, the best we can do is compute the perplexity of the training set, perform human evaluations, and provide samples for readers to judge. Figure 4 was intended as an evaluation of timing alone—it makes no claims about distance in content from the ground truth, and we do not perform any kind of alignment. It demonstrates simply that the model has learned a realistic distribution of message timings. Given that the model is only ever exposed to timing information in text form, this is quite nontrivial.
>
> **Re. Q3**
>
> This is a good question, and one we should have answered better in the original manuscript. The main difference (and an advance of the instant messaging evaluation) is that the human rater participated in the original instant messenger chat history, so they were much more qualified to judge how well the model mimicked the participants (i.e., fidelity). For the court transcripts, they were not intimately familiar with the individual judges or their jurisprudence, so we had to rely on more general ratings.

---

> > ### Author Response · Authors · 2024-11-23
> >
> > We hope we have addressed most of the reviewer’s concerns and would like to thank them again for their time. Please let us know if they’d consider recommending acceptance.

---

### Official Review · Reviewer_6JmG · 2024-11-04

**Soundness:** 3
**Presentation:** 2
**Contribution:** 3
**Rating:** 5
**Confidence:** 2

**Summary:**

The paper presents a method for simulating real-time interactive conversations by modeling diarized, timed transcripts, combined with causal rejection sampling. This method enables pre-trained, text-only language models to handle asynchronous and synchronous dialogues, as demonstrated with case studies involving instant messaging and spoken conversation simulations. The approach aims to maintain interactivity with minimal hardware requirements, while retaining a natural flow of dialogue.

**Strengths:**

1. Introduces a feasible approach for real-time interaction modeling using timed diarized transcripts and causal rejection sampling, which can be integrated into standard pre-trained models.
2. Demonstrates applicability with two distinct real-world cases: asynchronous instant messaging and real-time spoken conversations, adding diversity to potential model interactions.

**Weaknesses:**

1. There is little comparison with already existing work in this area (i.e. https://arxiv.org/abs/2405.19487)
2. The model is trained on narrow datasets (instant messaging and court transcripts), raising doubts about its generalization to diverse, real-world conversational scenarios.
3. Reliance on ASR and TTS systems may introduce errors and disrupt natural flow in spoken dialogues. An end-to-end audio model could reduce these issues by handling speech inputs and outputs directly.

**Questions:**

1. What considerations were made regarding the potential for user fatigue in interactions involving high rejection rates in real-time conversations?
2. Have you considered building an end-to-end model? Integrating ASR and TTS into the LLM as well to achieve faster response times?

---

> ### Author Response · Authors · 2024-11-23
>
> We thank the reviewer for their time. We address their individual concerns below:
>
> **Re. related work**
>
> Thanks for bringing this to our attention. While we disagree that we do not engage with related work more generally, we will incorporate it and related papers into our related work section. We note that a) our method has clear advantages: it accommodates multiple participants (not just two) and does not require any external machinery (like the state machine employed here) and b) accommodates arbitrarily long gaps between messages, simulating realistic interactions over time (e.g. conversations resumed on the following day). We also note that this work is concurrent with ours, which was publicly preprinted in May of this year.
>
> **Re. dataset choice**
>
> We are not aware of diverse, real-world timed dialogue datasets that are publicly available.. Nevertheless, we evaluated the models in both asynchronous and synchronous settings and found that they performed well in both. We have no reason to suspect the method would fail on other conversational datasets or wider domains, since it is generic and leverages pretrained LLMs.
>
> **Re. user fatigue**
>
> Our goal in this paper was simply to model realistic conversations, not to improve user retention. We have no evidence that our models output unrealistically tedious dialogue: in our quantitative and qualitative evaluations, our best models quite closely approximated human timings and content. While they certainly fell short in some regards, which we highlight in the paper, high rejection rates were not a problem we observed.
>
> **Re. end-to-end generation**
>
> This is an interesting idea, and would be a good direction for future work. We decided not to focus on this for this paper because we wished to illustrate the generality of our approach, which can be applied to any pretrained language model with minimal modifications. Second, there is simply a smaller variety of open-source text-audio models to experiment with. We expect that additional work on datasets, architectural modifications, modalities, etc. could be used to improve specific applications
>
> Please let us know if the reviewer has any outstanding concerns and whether they'd be willing to raise their score.

---

### Official Review · Reviewer_4VTs · 2024-11-06

**Soundness:** 3
**Presentation:** 3
**Contribution:** 3
**Rating:** 8
**Confidence:** 3

**Summary:**

The paper describes a method for applying a standard LLM to the task of generating dialogues incrementally, either turn-by-turn or word-by-word, by considering an output as a triple of (utterance time, speaker, message) and predicting on that basis. Experiments are given with turn-by-turn text-based instant messaging, and word-by-word text-based dialogue (starting with spoken dialogue, but processing with ASR to a text transcript and then experimenting on that). Experiments show that some plausible outputs can be generated, that generally bigger LLMs outperform smaller ones, and that the better models can receive good ratings from human evaluators.

**Strengths:**

The paper deals with interactive online generation of dialogue, in both message-by-message and word-by-word scenarios; this is a really important task from the point of view of building genuinely interactive conversational agents, and one that really hasn't seen much attention recently. The approach taken is quite intuitive to understand and is explained quite clearly. The evaluation shows that the approach can be effective, not only in generating coherent dialogue turns but including some important dialogue phenomena such as self-repair and turn-taking phrases (as shown by the examples in the appendix).

**Weaknesses:**

There is a fairly large amount of pre-LLM work on the details of incremental word-by-word dialogue modelling, including agents that can listen and generate on a word-by-word basis: the paper cites Skantze (2021)'s review of turn-taking treatment but would be stronger if it gave more comparison to work in that area.

The evaluation gives comparisons between LLMs on general metrics, but could be stronger if it looked at more details of coherence, turn-taking etc. It would be really nice if there was some discussion (quantitative or qualitative) of the linguistic phenomena that do/don't seem to be captured by this approach: the paper mentions coherence and consistency, but word-by-word and turn-by-turn modelling mean that other issues become important, e.g. realistic turn-taking patterns, interruption and overlap. The appendix transcripts show that some cases display good examples of some of these, some less so.

The evaluation uses data with a fairly low level of interactivity compared to many dialogue datasets. Instant messaging is an asynchronous, turn-by-turn medium; the spoken dialogues must be treated more incrementally, and are modelled word-by-word, but come from court transcripts in which the level of formality is high, and thus speaker changes, overlaps, interruptions etc are likely to be rare compared to more everyday, informal conversation. It would be helpful to know more about the average turn length, turn duration, number of speaker changes etc in this data compared to some of the more standard conversational datasets used in dialogue system development.

The model is one of generating a transcript as a whole, including all participants, whereas a practical conversational agent would have to react to other agents' contributions, online, and continue to adapt as their contributions come in (possibly interrupting, overlapping, leading to conversational directions that are not in the agent's interest, etc) - so some discussion of what would be involved in fitting this approach into those kind of constraints would be very helpful.

Relatedly, the model's basic assumption that p(e|context) can be decomposed into p(t|context) x p(s|context,t) x [...] seems to mean that it's predicting an utterance event time, and then predicting the speaker of that utterance event. A more realistic agent might predict speaker (or know that it wants to speak) first, and then need to predict when to speak - would that fit with this approach?

**Questions:**

See weaknesses above

---

> ### Author Response · Authors · 2024-11-23
>
> We are glad the reviewer agrees on the importance of the task and that they find our approach intuitive. We address their concerns below:
>
> **Re. related work**
>
> Gladly! We will expand this paragraph of the related work in the final version of the manuscript.
>
> **Re. metadata**
>
> The reviewer is correct that important metadata from our datasets, like turn length, are missing. We have included this information in the most recent version of the manuscript. We note that, while the court data is indeed formal, the chat data is highly informal, and includes many interruptions and rapid turn changes. The best models were able to handle both registers well.
>
> **Re. additional metrics**
>
> While we did not factor out metrics like how realistic turn taking patterns were, these considerations did play a role in “consistency” scores. We also do include qualitative analysis of the chats in Section 3.1.2.
>
> **Re. multi-agent conversations**
>
> In principle, our method should be able to handle the sort of situation you describe natively, as long as the training data includes examples of boisterous, interruption-prone multi-agent dialogue and overlapping dialogue is transcribed reasonably (though too much overlapping dialogue would raise minimum token generation rates for the LLM). Note that a model trained on multi-agent transcripts can “play” as many or as few of the participants as desired; we can prevent it from generating messages “from” another agent by modifying the sampling procedure. This is what we did for the interactive demos. One of the main strengths of our approach, in our opinion, is its generality.
>
> **Re. order of timestamp and speaker ID**
>
> Good observation! Our initial experiments did indeed put speaker IDs first, before the timestamp, and our paper was titled “... diarized, timed transcripts” instead of “... timed, diarized transcripts.” While both approaches produce realistic dialogues, putting the timestamp first is desirable for complicated technical reasons (specifically, to facilitate our speculative decoding procedure). Otherwise, the difference is fairly philosophical.
>
> We thank the reviewer for their time and hope we have addressed their concerns.

---

### Meta-Review · Area_Chair_ZV7w · 2024-12-20

**Metareview:**

This paper presents an interesting method for creating transcripts, however reviewers felt that the work was not well positioned with respect to other existing work in the area (references have been given by individual reviewers) and that the method was only tested on very narrow datasets: conversations between the two first authors and a subset of court cases of the US Supreme Court.  The evaluation was seen as lacking in detail, described as: "the number of conversations evaluated per method, the evaluation rubric, and consistency across ratings."  There were also questions raised as to whether or not the method could handle interruptions in a timely manner.

Reviewers think that the paper has promise but that a re-write with better positioning and more extensive evaluation are needed (including evaluating how the system performs under interruption).

**Additional Comments On Reviewer Discussion:**

Unfortunately, reviewers' discussion was low.  No reviewer responded to authors rebuttals.  The scores were 8,5,5, and 3.  Surprisingly, the person who rated the paper "8" had an extensive list of criticisms that were mirrored  by other reviewers who gave the paper far lower scores.  I read the paper as well and agree with many of the criticisms raised.  The paper does not adequately present prior work in transcript generation.  The training and testing are done on "narrow" datasets - the authors own chat history (not being disclosed, they say you can try it on your own) and a random subsampling of transcripts fro the US supreme court cases.

In summary, the reviewers pointed out many weaknesses of the paper but unfortunately did not return for discussion which is an undesirable outcome.  I have read the reviewers responses and the paper and overall my recommendation is based on the fact that 3 out of four reviewers recommend reject and the reviewer who said "accept" seemed to do so despite pointing out many of the same weaknesses.  I think it is clear what the authors need to do to improve the paper.

---

### Decision · Program_Chairs · 2025-01-22

Reject